# Ex Vivo Fluorescence Confocal Microscopy of MRI-Guided Targeted Prostate Biopsies for Rapid Detection of Clinically Significant Carcinomas—A Feasibility Study

**DOI:** 10.3390/cancers16050873

**Published:** 2024-02-22

**Authors:** Ulf Titze, Barbara Titze, Torsten Hansen, Peter J. Barth, Furat Abd Ali, Fried Schneider, Matthias Benndorf, Karl-Dietrich Sievert

**Affiliations:** 1Department of Pathology, Medical School and University Medical Center OWL, Klinikum Lippe Detmold, Bielefeld University, 32756 Detmold, Germany; barbara.titze@klinikum-lippe.de (B.T.); t.hansen@patho-trier.de (T.H.); 2MVZ for Histology, Cytology and Molecular Diagnostics Trier GmbH, 54296 Trier, Germany; 3Gerhard-Domagk-Institute of Pathology, Münster University Hospital, University of Münster, 48149 Münster, Germany; peter.barth@ukmuenster.de; 4Department of Urology, Medical School and University Medical Center OWL, Klinikum Lippe Detmold, Bielefeld University, 32756 Detmold, Germany; furat.abdali@klinikum-lippe.de (F.A.A.); fried.schneider@klinikum-lippe.de (F.S.); karl-dietrich.sievert@klinikum-lippe.de (K.-D.S.); 5Department of Diagnostic and Interventional Radiology, Medical School and University Medical Center OWL, Klinikum Lippe Detmold, Bielefeld University, 32756 Detmold, Germany; matthias.benndorf@klinikum-lippe.de

**Keywords:** confocal microscopy, prostate cancer, prostate MRI, targeted prostate biopsy

## Abstract

**Simple Summary:**

Ex vivo fluorescence confocal microscopy is a novel microscopic technique that allows lossless analysis of unfixed tissue. By examining MRI-guided biopsies from suspicious lesions of the prostate, the majority of prostate carcinomas requiring treatment can be diagnosed. The recorded digital images are a promising basis for future AI-supported diagnostic algorithms. The rapid feedback within a few hours of the biopsy collection can reduce the stress burden on patients and contribute to better patient care, as the necessary steps for the therapies can be organized promptly. It is also a promising diagnostic approach for future local therapies in which effective substances can be applied specifically to tumor foci.

**Abstract:**

Background: MRI-guided prostate biopsies from visible tumor-specific lesions (TBx) can be used to diagnose clinically significant carcinomas (csPCa) requiring treatment more selectively than conventional systematic biopsies (SBx). Ex vivo fluorescence confocal microscopy (FCM) is a novel technique that can be used to examine TBx prior to conventional histologic workup. Methods: TBx from 150 patients were examined with FCM on the day of collection. Preliminary findings were reported within 2 h of collection. The results were statistically compared with the final histology. Results: 27/40 (68%) of the csPCa were already recognized in the intraday FCM in accordance with the results of conventional histology. Even non-significant carcinomas (cisPCa) of the intermediate and high-risk groups (serum prostate-specific antigen (PSA) > 10 or 20 ng/mL) according to conventional risk stratifications were reliably detectable. In contrast, small foci of cisPCa were often not detected or were difficult to distinguish from reactive changes. Conclusion: The rapid reporting of preliminary FCM findings helps to reduce the psychological stress on patients, and can improve the clinical management of csPCa. Additional SBx can be avoided in individual cases, leading to lower rates of complications and scarring in the future surgical area. Additional staging examinations can be arranged without losing time. FCM represents a promising basis for future AI-based diagnostic algorithms.

## 1. Introduction

Prostate cancer (PCa) is the most commonly diagnosed cancer and the second most common cause of cancer death among men in the United States [1]. It is long known to be a morphologically, genetically, and clinically heterogeneous tumor disease [2]. The clinical courses vary from slow-growing tumors in old age to highly aggressive metastatic tumors in young men. The vast majority of tumors are diagnosed at an advanced age.

PSA screening has contributed significantly to a reduction in mortality from PCa [3], but carries the risks of overdiagnosis and overtreatment [4]. In recent years, there has been a paradigm shift in the diagnosis of prostate carcinoma with the aim of filtering out indolent prostate carcinomas and diagnosing tumors leading to the death of affected patients as selectively as possible. Magnetic resonance imaging (MRI) has led to a paradigm shift in the diagnosis of prostate cancer. Radiologists use the standardized Prostate Imaging Reporting and Data System (PI-RADS) to communicate the degree of suspicion for prostate cancer of visible lesions in the glandular parenchyma [5]. A low PI-RADS score (1 or 2) suggests a low likelihood of clinically significant cancer, while a high PI-RADS score (4 or 5) indicates a higher likelihood of cancer, with 5 being the most suspicious category. Furthermore MRI imaging of the prostate enables targeted biopsies of individual foci [6,7].

Conventional risk stratification systems consider PSA values in the serum and the histological results, including the Gleason score (GS) and the presumed size of tumors based on their presentation in the biopsies. In contrast, most studies of MRI-based diagnostics define clinically significant prostate carcinomas (csPCa) as GS ≥ 7a tumors, while GS ≤ 6 carcinomas are classified as clinically insignificant (cisPCa) regardless of the PSA values [8,9,10,11,12]. Performing MRI-guided biopsies (TBx) of suspicious lesions leads to higher detection rates of csPCa and to less cisPCa than taking systematic biopsies (SBx). One result of the GÖTEBORG-2 study was that the omission of SBx halved the detection of cisPCa, but also missed one in five csPCa [13].

Several autopsy studies have shown a high incidence (36–51%) of undiagnosed csPCa, which did not limit life expectancy, but would have been assessed as requiring treatment if diagnosed [14]. These findings justify the cautious diagnostic strategy, which visibly accepts the overlooking of cisPCa and some csPCa, since, in contrast to other organ tumors, tumor progression cannot be assumed due to delayed diagnosis.

Several studies have investigated the pyscho-social consequences of the diagnosis and treatment of prostate carcinoma [15,16]. In a 2005 study, it was shown that the patients’ anxiety levels were among the highest in the time interval between the first suspicion of prostate cancer and the receipt of the biopsy result [17]. Individual publications give this waiting time as a median of 53 days (or 7.7 weeks, respectively) [18,19].

Ex vivo fluorescence microscopy (FCM) is a novel microimaging technique that enables rapid histologic examinations of native biopsies from the prostate and other organs [20,21]. In this prospective study, MRI-guided prostate biopsies from visible lesions of the prostate (TBx) were examined with FCM within 2 h after acquisition. The aim was to investigate to what extent FCM analyses of TBx are suitable for detecting csPCa and predicting the outcome of tumor boards on the day of biopsy collection.

## 2. Materials and Methods

### 2.1. Study Design

The aim of this study was to compare the intraday diagnoses from the FCM analysis of TBx with the final treatment recommendation of the tumor board. The TBx were first placed in saline solution in the operating room after collection, whereas the SBx were immediately fixed in formalin. All sample tubes were brought to the Institute of Pathology by a transport service. The TBx were examined with FCM immediately after receipt and the biopsy result was reported within 2 h. The FCM scans were second reviewed by another experienced pathologist (BT) with specialization outside uropathology. Furthermore, the histological slides of the FFPE-processed material (TBx and SBx) were subjected to a second review by an external uropathologist (PJB) in order to validate the extent and grading of the tumors recorded.

The first endpoint was the level of agreement between the results of FCM analysis and the final conventional histologic examination of the TBx. The second endpoint was the extent to which timely FCM analysis of the TBx could predict treatment decisions of the tumor board based on PSA and histological examinations of both TBx and SBx. Furthermore, the interobserver agreement for the assessment of FCM scans was analyzed. Figure 1 provides an overview of the study design.

### 2.2. Patients

The cohort comprised 150 men aged between 42 and 86 years (mean 67.5 ± 8.7 years) who underwent prostate biopsy in the University Medical Center OWL, Germany, Campus Lippe, Department of Urology, as part of clinical care. The indications for the biopsy procedures were clinical suspicion of prostate cancer due to elevated serum PSA levels and/or tumor suspicious findings on imaging and/or clinical examination (n = 133) or follow-up biopsies for known prostate cancer under active surveillance (n = 17 patients). Complete MRI imaging and current PSA values were available for all patients at the time of the procedure. The detailed clinical data have been published elsewhere [22], whereas the data of our intraoperative FCM-examinations of the TBx has not been published before for this cohort. Written informed consent to participate in the study was obtained from all patients. All trials were conducted in accordance with the provisions of the Helsinki Declaration and were approved by the local ethics committee.

### 2.3. Biopsy Procedure

PI-RADS category 3–5 lesions in the MRI data were reconstructed in three dimensions (contouring) prior to the biopsy procedure. The MRI images were fused with real-time ultrasonography using software (BK-5000 ultrasound system, BK medical, Burlington, MA, USA; BioJetTM MRI/ultrasound fusion system, Medical Targeting Technologies GmbH, Barum, Germany). This system was used to take targeted biopsies from the conspicuous foci (target biopsies, TBx) and systematically from the inconspicuous parenchyma of the remaining sectors (system biopsies, SBx) using 3D imaging. A fully automated biopsy system (DeltaCut, PAJUNK GmbH, Geisingen, Germany) was available for the biopsy procedure. Detailed information on the biopsy procedure can be found in our previous publication [22].

### 2.4. Specimen Handling and Ex Vivo Confocal Microscopy of the Target Biopsies

Biopsies were processed at the Institute of Pathology within one hour of receipt. The unfixed TBx were pretreated in 70% ethanol for 10 s and stained in 0.4% acridine orange solution. The so dyed native biopsies were aligned on a special slide, gently spread with a soft embedding sponge and examined with a VivaScope G4. The digital scans were stored pseudonymously and promptly reviewed by an experienced uropathologist (UT). The scans were screened for the presence of prostate carcinoma. The detected prostate carcinomas were graded according to the Gleason grading system. Furthermore, the tumor extension was documented as an absolute length in mm and as degree of infiltration of the entire biopsy cylinder.

Subsequently, the native TBx were transferred into buffered formalin and processed together with the SBx according to the standardized procedure for formalin-fixed and paraffin-embedded (FFPE) material. The histological diagnosis was carried out by two experienced uropathologists (TH, UT) as part of routine care in accordance with current guidelines. In cases with difficult dignity assessment, the standard stains (hematoxylin & eosin) were supplemented by immunohistological examinations (p40, AMACR). The indication for interventional therapies was determined in an interdisciplinary tumor board on the basis of the final histological examination of TBx and SBx, current PSA value, MRI findings, age and family history.

### 2.5. Intraoperative FCM Analysis of the Target Biopsies as a Triage Test

The FCM analysis of the TBx took place immediately before the histological examination of the biopsy material as a loss-free intermediate step. The FCM was used to examine the TBx for the presence of tumors. The system software enabled a quantitative measurement of tumor foci in µm and as a percentage of the length of the entire biopsy cylinder (infiltration grade). Furthermore, the tumors were graded according to their histological growth pattern and assigned to the revised Gleason groups. In patients with two or more lesions, the FCM results of the TBx were summarized and the highest GS and infiltration grades were recorded.

At the time of the FCM examination, the patient’s age, family history and PSA value were available for risk stratification. Patients with GS ≥ 7 tumors were classified as csPCa. Additionally, patients with GS6 carcinomas in the FCM scans (cisPCa) were grouped according to the National Comprehensive Cancer Network (NCCN) Prostate Cancer risk Classification in synopsis with the PSA values and the measured infiltration levels [23]. Here, tumors in the low-, intermediate- or high-risk group are also classified as potentially requiring treatment, whereas tumors of the low-risk group are classified feasible for active surveillance . The results of both risk stratification systems (csPCa vs. cisPCa and NCCN) were separately analyzed in the further evaluations.

These preliminary results were communicated to the treating urologist by telephone within 2 h of the biopsy being taken. In a joint discussion, the patients were grouped into 4 preliminary categories: 1. patients without tumor detection in the TBx; 2. patients with suspicious glands in TBx; 3. patients with clinically non-significant prostate cancer and 4. patients with clinically significant prostate cancer. At the patients’ request, they were informed of the preliminary results before being dismissed from hospital.

### 2.6. Statistical Analyses

The FCM diagnoses were transferred to an ordinal scale for the statistical analyses (0—no tumor, 1—suspected tumor, 2—prostate carcinoma without need for therapy, 3—PCa requiring therapy). Since immunohistochemical methods were also available for the final histological work-up, “suspected prostate carcinoma” was not a diagnostic option here. The results were therefore transferred to a tripartite diagnostic scale: 0—no tumor, 1—PCa without need for therapy and 2—PCa requiring therapy. To achieve tripartite scale systems for the statistical comparison of both diagnostic procedures, the values 0 (no tumor) and 1 (suspected tumor) were combined as a common category without definitive tumor detection for the FCM-ratings.

The results of the FCM analyses and the final histology (TBx alone or TBx + SBx) were plotted in error matrices. Sensitivity and specificity, as well as the positive and negative predictive values (PPV and NPV) were calculated for the detection of carcinomas in total and for carcinomas requiring treatment in special. The levels of agreement between the diagnostic procedures and between the observers were determined using Cohen’s Kappa and evaluated according to the criteria of Landis/Koch [24,25].

## 3. Results

### 3.1. Final Tumor Diagnoses in the Cohort

In the synopsis of all biopsies (TBx + SBx), prostate carcinoma was detectable in 96 patients. 56 cases were GS6 carcinomas in the final histology, 33 cases were GS7a carcinomas and 7 cases were GS7b or higher. Please see our previous publication for further details of the tumor diagnoses [22].

### 3.2. MRI-Findings and Detection of Cancer in the TBx

The MRI data of 150 patients showed 215 tumor-suspected foci in total (1.4 ± 0.7 lesions per patient, range 1–4 lesions). These were 54 PI-RADS group 5 lesions, 121 PI-RADS group 4 lesions, 30 PI-RADS group 3 lesions and 8 PI-RADS groups 2 lesions. Two lesions were not documented consistently. Depending on their sizes, 1–8 biopsies (mean 3.3 ± 1.2) were taken per lesion. Prostate carcinoma was detected in 88/215 foci (76 patients). Conventional histology of the TBx revealed GS6-patterns in 45 cases, GS7a-patterns in 26 cases and GS7b-5 patterns were visible in 5 cases. Please see our previous publication for further details of the MRI findings [22].

The detection rates for prostate cancer increased with their increasing Gleason grade. Of 56 men with GS6 tumors in the cohort, 40 (71%) were detected in the TBx and 16 tumors (29%) were missed. In total, 29/33 (88%) patients with GS7a-PCa in the final histology showed tumor foci in the TBx. However, in 5 of these cases, only GS6 patterns were present in the TBx which led to an underdiagnosis. Another four GS7a carcinoma were missed in the TBx. Representative tumor foci of all GS7b-10 carcinomas were visible in the TBx of the affected patients (n = 7). Two of these cases showed only GS7a patterns in the TBx but no low grade tumor lesions.

### 3.3. Detection of Cancer in the Intraday FCM-Examination of the TBx

Altogether, 30/40 of the GS ≤ 6 tumors (cisPCa) present in the TBx were correctly identified and graded in intraoperative FCM. Focal tumor infiltrates were only addressed as suspicious for malignancy in 6 of these cases and missed in 4 cases. The missed tumors were small foci (mean size 1.25 mm, range: 0.6–2.2 mm) that were difficult to assess even by conventional histology and could only be reliably classified as malignant by immunohistology. The tumors that were assessed as suspicious in the FCM were little larger (mean foci size 2.5 mm, range 0.8–4 mm). In these cases, the presented glands were well differentiated and difficult to distinguish from reactive changes. In some of these cases, limited scan quality led to incomplete visualization of the histoarchitecture and cytological criteria. Thus, larger foci (mean foci size 2.5 mm, range 0.8–4 mm) of well differentiated glands could not be reliably differentiated from benign glandular changes in these images.

The key to the diagnosis of csPCa was the detection of Gleason 4 and 5 patterns in the histologic sections or FCM scans. The FCM-diagnosis of Gleason 4 patterns was challenging in some cases. Since its first definition in 1966, four basic histologic growth patterns have been defined in several consensus conferences: poorly formed glands, fused glands, cribriform structures and glomerular structures. All of these growth patterns were represented in the TBx an were recognizable in the FCM scans (Figure 2). As in conventional histology, there were interrater discrepancies in the diagnosis of the poorly formed glands and fused glands patterns, which in some cases were only focal. These patterns had been controversially discussed in the earlier consensus classifications due to their subtle overlap with Gleason 3 patterns, and continue to be the most common cause for second opinions. In contrast, the detection of glomeruloid and cribriform patterns did not pose a major difficulty when present on FCM scans. The detection of Gleason 5 patterns was more challenging in the FCM scans than in conventional histology, as poorly differentiated tumor cell in the unfixed tissue show similarities to unfixed stromal cells and non-cohesive tumor cell clusters were difficult to distinguish from stromal nodules.

Of 29 cases in which GS7a carcinomas were recorded in the TBx, 27 (93%) were recognized as malignant in the FCM analysis and 2 cases were classified as suspicious for malignancy. 24 of the cases of this group showed GLEASON-4 patterns in the TBx (Figure 3), which were correctly diagnosed in the FCM in 20 cases (83%). In 3 cases, the high-grade patterns were not clearly recognizable in the FCM scans due to sampling errors and were undergraded as GS6 tumors. The high-grade component was only diagnosed with certainty in conventional histology. One of these cases could only be diagnosed as suspicious for malignancy in these limited digital scans.

Five cases from the group of GS7a tumors showed exclusively GS6 patterns in the TBx and were up-graded by the SBx (Figure 4). Three of these cases showed foci sizes of <5 mm, the other two case showed tumor sizes of 9.2 mm and 9.5 mm (mean 5.5 ± 3.6 mm, range 2.0–9.5 mm). The focal sizes of these cases did not differ statistically from the other cases of GS6 carcinoma the TBx and there was no statistical difference in the tumor sizes of both groups (mean focal size of the GS6 tumors in the TBx: 5.5 ± 3.8 mm, range 0.2–15 mm, *p* = 0.48).

All GS7b-10 tumors (n = 7 cases) were present in the TBx, mostly as large foci (mean size 9.0 ± 5.7 mm, Range 2.5–16 mm). 2 cases showed GS7a patterns only but still were recognizable as high grade cancer. The tumors were clearly recognizable in the FCM scans and were correctly diagnosed as high-grade carcinomas in the intraday examination.

In summary, 64/76 of the tumors present in the TBx were reliably detected in the FCM (sensitivity 84%, specificity 100%, NPV 86%, kappa 0.84). In total, 27/31 tumors with high-grade morphology were correctly classified as clinically significant carcinomas (sensitivity 87%, specificity 100%, NPV 97%, kappa 0.91).

### 3.4. Detection Rate of csPCa in the Intraday FCM-Examination of the TBx

The detection rate of carcinomas requiring treatment depended on the definition of clinically significant prostate carcinomas (csPCa). In more recent studies, csPCa were predominantly defined as GS ≥ 7 carcinomas, whereas GS ≤ 6 carcinomas are classified as clinically insignificant (cisPCa) regardless of the PSA levels in serum and their sizes or extent in the prostate biopsies. The results are shown in Table 1.

The patient cohort comprised 40/96 patients with csPCa (33 GS7a and 7 GS7b-10). Only 31/40 cases were recognizable as high-grade carcinomas in the TBx. As discussed above, only the GS6 component was present in the TBx in 5/40 cases leading to their undegrading as cisPCa. In 4/40 cases, no tumor was detectable in the TBx and only present in the SBx. The FCM analysis of the TBx correctly diagnosed 27/40 of the csPCa of the cohort (sensitivity 68%, specificity 100%, N.p.v. 89%, kappa 0.75). Recognizable GLEASON-4-patterns present in the TBx were not recognized in the FCM scans in 3 cases and one of these tumors was missed due to an insufficient image quality. In summary, a total of 8/40 (20%) of the csPCA were under-graded as cisPCa in the FCM. As mentioned above, one case presenting with GS6 tumor in the TBx exclusively was classified as suspicious for tumor.

In the conventional systems of the currently valid guidelines, risk stratification is based on serology (PSA < 10, 10–20 oder > 20) and the tumor size estimated on the basis of the biopsy result (GS, numbers of positive cores and percentage of the cores involved). GL ≤ 6 carcinomas in the very-low-risk group (PSA ≤ 10 ng/mL and tumor in ≤2 biopsies with an infiltration grade > 50%) could still be actively monitored, while interventional therapies (prostatectomy, radiation therapy) were recommended for low-, intermediate- and high-risk tumors [23,26,27].

Our cohort comprised 56 patients with GS6 carcinomas. Based on the serum PSA level and biopsy extension, 26/56 (46%) patients were assigned to the very low-risk group, 13/56 (23%) to the low-risk group, 14/56 (25%) to the intermediate-risk group and 3/56 (5%) to the high-risk group (Figure 5). In this way, 30 additional patients of the total collective with ISUP grade 1 tumors of the low-risk to high-risk group were classified as requiring treatment.

In total, 15/26 (58%) of the very-low-risk tumors were detected in the TBx, presenting with larger foci than in the SBx (TBx: mean 4.4 ± 3.1, Range 0.6–9.9 mm vs. SBx: mean 1.2 ± 0.9 mm, range 0.1–2.7 mm). Only about one third (9/26, 34.6%) of the cases in this group were diagnosed in the FCM survey. A further third (8/26, 30.8%) of the cases were assessed as suspicious and the last third (9/26, 34.6%) were not recorded.

The tumors in the low-risk group posed similar difficulties. Only 3/13 (23%) of the tumors in this group were correctly classified in the intraday examination (mean 10.3 ± 4.8 mm, range 5.5–15.0 mm, mean percentage 65 ± 5.0%). The majority of the cases in this group were provisionally assigned to the very low-risk group due to their small sizes (mean 8.3 ± 3.7 mm, range 1.8–12.1 mm, mean percentage 38 ± 24%) and low PSA values and could only be correctly classified in conjunction with the findings of the SBx. A total of four of the tumors (31%) in this group were only addressed as suspicious or missed.

The GS6 tumors in the intermediate- and high-risk group were identified solely on the basis of the PSA value (PSA > 10 or >20, respectively). There were no significant differences to the tumors in the low-risk groups with regard to their focal size in the TBx (intermediate risk tumors: mean 4.2 ± 3.8 mm, range 0.1–10.2 mm; high-risk tumors mean 6.8 ± 0.9 mm, range 6.1–7.9 mm). The synopsis of positive tumor detection and elevated PSA values resulted in classifications that were consistent with the tumor board in all cases of these risk groups.

In summary (Table 1), 70 of the 96 men with PCa were indicated for interventional therapy on the basis of conventional NCCN-risk stratification. Of these, 46 patients were correctly identified by the intraday FCM analysis of TBx (sensitivity 66%, spec 100%, NPV 77%, kappa 0.67).

### 3.5. FCM Analysis of Target Biopsies as an Intraday Triage Test

In 66 men, the TBx were assessed as tumor-free in the FCM triage test (Table 2). This finding was confirmed in 46/66 (70%) cases. In 20 (30%) of the patients, a tumor was detected in the conventional histological work-up. Of these, 16 cases were GS6 carcinomas corresponding to cisPCa and 4/20 csPCa (GS7a). Based on conventional risk stratification, the patients were divided into 9 tumors in the very-low-risk group and 11 tumors in the low- to high-risk group with indication for interventional therapy.

In 20 cases, suspicious glands were conspicuous in the FCM. In 12 of these cases, the atypical glands were found to be PCa on FFPE histology, 10 of them with GS6 tumors and 2 cases with GS7 (csPCa). 8 tumors belonged to the very-low-risk group, 4 lesions were tumors in need of intervention in the low- and intermediate-risk group.

Altogether, 18 men with PSA < 10 were diagnosed with GS6 carcinoma in the FCM analysis without a definite need for treatment. Of these, nine were found to be GS6 carcinomas in the very-low-risk group for active surveillance. Six cases proved to be low-risk tumors in the histologic workup with indication for interventional therapy according to the current guidelines. Three cases were upgraded to GS7a- carcinoma, corresponding to an intermediate risk disease.

PCa requiring treatment was intraoperatively diagnosed in 46 patients. Consistent with the final histology, these were 31 GS > 7 tumors and 15 GS6 tumors of the intermediate- and high-risk groups (12 cases with PSA > 10 and 3 cases with an infiltration grade >50%).

In summary, every FCM diagnosis of a carcinoma requiring treatment (depending on the cut-off for GS7 or low-risk PCa) was confirmed in the final histology (specificity 100%). In case of negative tumor findings, a tumor diagnosis was to be expected in 30% post hoc, but most of these did not require treatment. The majority of atypical glands also corresponded to PCa that did not require treatment. The main problem was the classification of GS6 carcinomas, partly due to post hoc upgrade or the discrepant risk stratification.

### 3.6. Interobserver Analysis of FCM Diagnoses

Table 3 shows the field chart of the two FCM examiners (UT intraoperative and BT post hoc). 24/31 (77%) of the GS ≥ 7-Ca recorded in the TBx were consistently identified as csPCa by both examiners. Three cases were additionally correctly identified as GS7a by UT, while BT classified them as GS6. All GS7b-10 cases were consistently identified as csPCa. Similarly, 35/56 (63%) of GS6 tumors were consistently diagnosed and 2 additional cases were diagnosed by the specialist. There were clear differences in the evaluation of doubtful findings. Four cases were concordantly rated as suspicious by both raters, which finally proved to be three GS6 carcinomas and one GS7a carcinoma. UT classified 16 additional lesions as suspicious, of which 4 were found to be GS6 tumors and 12 were tumor-free. BT classified 3 additional lesions as suspicious, of which 2 were found to be GS6 tumors and 1 case was found to be tumor-free. Tumor-suspicious glands were over-diagnosed as GS6 carcinoma in 2 cases in the post hoc analysis by BT, which was not confirmed in the immunohistological evaluation.

Overall, substantial agreement was achieved between both investigators (kappa 0.78). The difference in specialization was noticeable in 5 cases (3× GS7a and 2× GS6, <10%).

## 4. Discussion

The prevalence of prostate cancer increases non-linearly with each decade of life, from <5% in men <30 years to 59% in men >79 years [28]. Several autopsy studies cited found a with age increasing reservoir of incidental PCa which did not limit life expectancy. PSA screening led to a reduction in mortality on the one hand [3], but involved a significant risk of overdiagnosis and overtreatment [29], which is why the US Preventive Services Task Force (USPSTF) continued to advise against PSA screening in males 70 years or older in 2018 and assumed only a small benefit for men aged 55–69 years [30,31]. Thus, PCa screening still is a matter of ongoing controversy and national and international guidelines have not reached a consensus [32].

The implementation of MRI in the diagnosis of prostate cancer represents a step towards the more selective diagnosis of PCa requiring treatment. PI-RADS grading allowed preselection of abnormal findings in the prostate, resulting in increased detection rates of csPCa [9]. Clinically significant PCa (GS > 7a) were detected more frequently with MRI-fused biopsies than with systematic biopsies [6]. Unfortunately, the biological potential of some of the csPCa was also underestimated by the isolated examination of TBx [33]—which may be tolerable in specific cases, provided the patients remain under observation.

However, the definition of PCa requiring intervention in the biopsy remains controversial. The majority of studies define csPCa as GS ≥ 7a carcinomas. The correct handling of large-volume GS-6 carcinomas is particularly problematic from a clinical perspective. These tumors are classified as cisPCa according to the cutoffs mentioned above. However, 80% of biopsy-diagnosed GS-6 lesions > 8 mm in size can be expected to upgrade to csPCa [34]. On the other hand, recent studies have shown that it is also reasonable to offer delayed treatment to some patients with intermediate-risk PCa [35]. However, there are currently no valid criteria for the selection of suitable patients [36]. From a practical point of view, in a significant proportion of cases, clinical, serological, and anamnestic parameters must still be used to make a final treatment recommendation, which leads to delayed decision-making.

We see the Ex Vivo FCM as another interesting tool that contributed to a more precise and individualized biopsy diagnosis of PCa due to the fast results. In this study, the preliminary results were transmitted to the urologist within two hours of biopsy collection. This was a comparatively simple method that could be used as an additional intermediate step before the actual histological processing. Our previous studies have already shown that the subsequent processing for conventional histology and, if necessary, immunohistological and molecular techniques were not impaired [37,38].

The visualization of the tissue in the digital scans was similar to hematoxylin-eosin-stained frozen sections of the native material. For methodological reasons, there were diagnostic limitations of FCM compared to conventional histology, particularly in the assessment of doubtful lesions. The nuclear staining by acridine orange was inhomogeneous in some biopsies, resulting in weakly stained sections in the FCM scans, in which the glandular architecture could only be assessed to a limited extent. In contrast, diagnostically useful artificial vacuolization of the cytoplasm of basal epithelia occurred on FFPE-processed slides, making these diagnostically important cells more easily recognizable. In addition, immunohistological techniques could be used in cases of doubt in conventional histology, which made it possible to conclusively characterize the remainder of doubtful findings. Small formations of abnormal glands could therefore only be assessed as malignant in the digital scans and only classified in conventional histology (with immunohistology if necessary). This led to some GS ≤ 6 carcinomas being overlooked or under-classified in the FCM, which were assigned to the low-risk and very-low-risk groups in the final histological assessment.

The reliable diagnosis of GLEASON-4 patterns is a therapeutically relevant step in the grading of prostate carcinoma, which also requires some experience in conventional histology [39] and often leads to a second opinion being obtained [40]. Naturally, there were also methodological limitations in the FCM findings, in which only a 0.5 µm thick optical section of the tissue was visualized. In contrast, the conventional histological examinations were based on at least 8 sections with a thickness of 4–10 µm, which enabled the evaluation of a larger volume of biopsies. Focal criteria of a GLEASON-4 pattern (glandular fusions, poorly formed glands or glomeruloid glands) were therefore more reliably recognizable in the FFPE-processed sections. This explains why some of the GS7a carcinomas were recognized as carcinomas in the preliminary triage test, but their degree of malignancy was underestimated. The analysis of the interobserver variability also revealed clear differences between the examiners depending on the level of experience.

Despite these methodological limitations, the FCM analysis of TBx was very well suited for the rapid detection of csPCa. The majority of tumors with a high-grade component were detected in accordance with the final histology and grouped as requiring treatment. In the group of GS ≤ 6-PCa, the tumors of the intermediate- and high-risk group were also reliably detected. The significant deviations between FCM and conventional histology within the GS ≤ 6-PCa appeared tolerable from a clinical point of view, as these tumors belonged to the low-risk and very-low-risk groups.

In this study design, ex vivo FCM of TBx was used as a preliminary triage test for the rapid diagnosis of csPCa prior to the actual histologic examination. A prerequisite was clear communication of the methodological limitations to the patients. With a preliminary negative result, cisPCa could be expected in one in four cases in the final result and csPCa in <10% of cases. The majority of the suspicious findings also corresponded to cisPCa. If cisPCa was diagnosed, an upgrade to csPCa was to be expected in one in five patients.

For patients, a prompt initial preliminary opinion by the treating physicians leads to a significant reduction of stress. Previous studies have shown that the anxiety level of men was highest in the period between the start of screening examinations and the biopsy diagnosis and decreased rapidly with the receipt of the biopsy result (interestingly independent of the diagnosis). According to the literature, the average waiting time for patients to receive their results is 53 days [41]. Prompt feedback on the result of the FCM analysis helps to ease the distressing uncertainty until the final biopsy result is received. The psychosocial effects of this additional diagnostic option are currently being systematically investigated in an interdisciplinary approach with urologists and psycho-oncologists. If csPCa is detected in the FCM analysis, the procedure offers patients the opportunity to take advantage of psycho-oncological services and contact self-help groups without losing too much time. The attending physicians can promptly initiate any necessary staging examinations.

Ex vivo FCM examinations can also be performed in the operating room in order to provide the examiners with feedback on the biopsied tissue within a few minutes. This results in the option of modifying the biopsy strategy during the session. If csPCa is detected in the TBx, the sampling of additional SBx can be avoided, which is expected to reduce perioperative complications. Less reparative inflammatory processes after the biopsy procedure would lead to less scarring, which would facilitate the exposure of the periprostatic connective tissue layers for nerve-sparing surgical techniques. Furthermore, the risk of nerve damage, often discussed as a result of the number of biopsies taken from critical areas like the apex of the prostate could be reduced [42]. Conversely, if TBx is negative or shows small insignificant cancer foci only, the decision over the acquisition of additional SBx can be made intraoperatively if a prostate carcinoma can still be assumed based on the clinical-serologic constellation or in case an upgrade to higher GLEASON grades is to be expected [43].

Due to its potential suitability for modifying intraoperative procedures, the ex vivo FCM analysis of TBx is also an interesting tool for future focal therapies. The targeted biopsy of one or more suspicious lesions from the prostate allows their dignity to be assessed intraoperatively. With the fusion of MRI and sonography, it is possible to target malignant lesions again and install therapeutically effective substances there. If necessary, the biopsy material is available without loss as fresh tissue for rapid molecular genetic tests.

These possibilities require urologists, radiologists and pathologists to work closely together in terms of time and location, which cannot be guaranteed in broad routine use at present. However, the combination of MRI and digital microscopic imaging represents an ideal basis for the development of neuronal networks. Promising results from recent studies suggest that AI-supported systems for lesion detection and PI-RADS scoring in MRI will represent further steps towards specific detection of csPCa [44]. The possibility of AI-supported diagnostics and web-based communication offers new perspectives for efficient interdisciplinary collaboration. 

## 5. Conclusions

FCM examinations of TBx from the prostate are a promising approach for the routine diagnosis of PCa to achieve a more effective clinical management of csPCa. The fast and reliable results of FCM can reduce the psychological burden on patients. It is to be expected that the prompt planning of subsequent diagnostic and therapeutic procedures will also contribute to a trusting relationship between patients and the centers treating them.

## Figures and Tables

**Figure 1 cancers-16-00873-f001:**
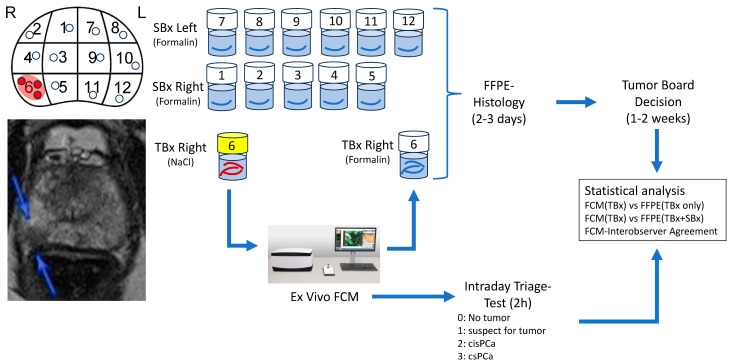
Study design. MRI-targeted biopsies (TBx) of visible lesions (arrows) as well as systematic biopsies (SBx) were collected from 150 men. TBx were sent to pathology in saline solution for an intraday triage test using ex vivo Fluorescence Confocal Microscopy (FCM). The TBx were FFPE-processed subsequently together with the SBx. The results of the intraday FCM-examination were compared to the results of the conventional histology of the FFPE-processed materials including immunohistology in doubtful cases.

**Figure 2 cancers-16-00873-f002:**
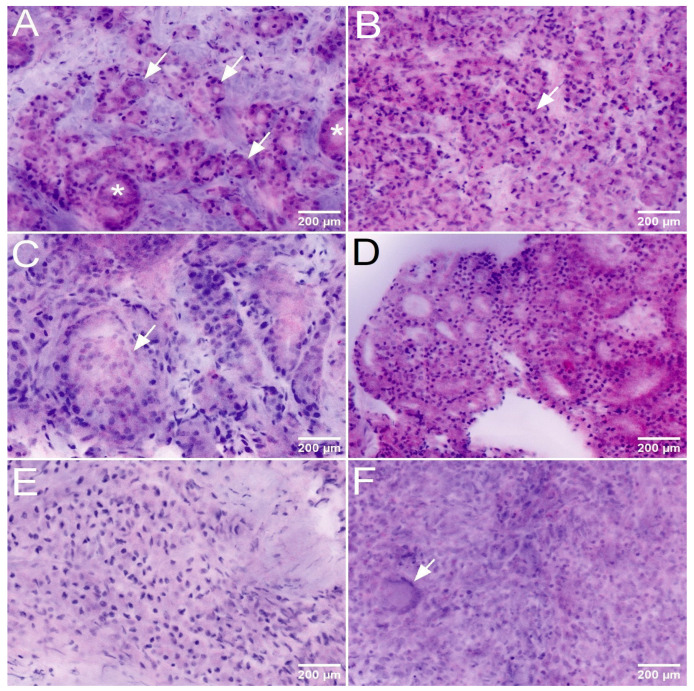
Presentation of csPCa and one of its mimickers in the FCM-Scans. (**A**–**D**): Gleason pattern 4 comprises a heterogenous group of tumors with 4 basic patterns. Poorly formed glands (**A**) were recognizable as >10 ill-defined glands with poorly formed glandular lumens (arrows). The size of the glands can be variable (*), which explains difficulties in the differentiation from Gleason 3 patterns. Fused glands (**B**) presented as composed groups of acini (arrow) that were no longer separated by stroma. (**C**) Glomeruloid (arrow) (**C**) structures were recognizable glands containing intraluminal cribriform structures with a single point of attachment, resembling a renal glomerulus. and cribriform structures (**D**). (**E**): Gleason pattern 5 corresponds to the least degree of glandular differentiation in prostatic acinar adenocarcinoma, defined as ‘very poorly differentiated tumors; usually solid masses or diffuse growth with little or no differentiation into glands’ in the original Gleason grading system. (**F**): Granulomatous prostatitis was difficult to distinguish from Gleason pattern 5 tumors in the FCM scans of one case. The reliable detection of giant cells (arrow) was an important diagnostic criterion.

**Figure 3 cancers-16-00873-f003:**
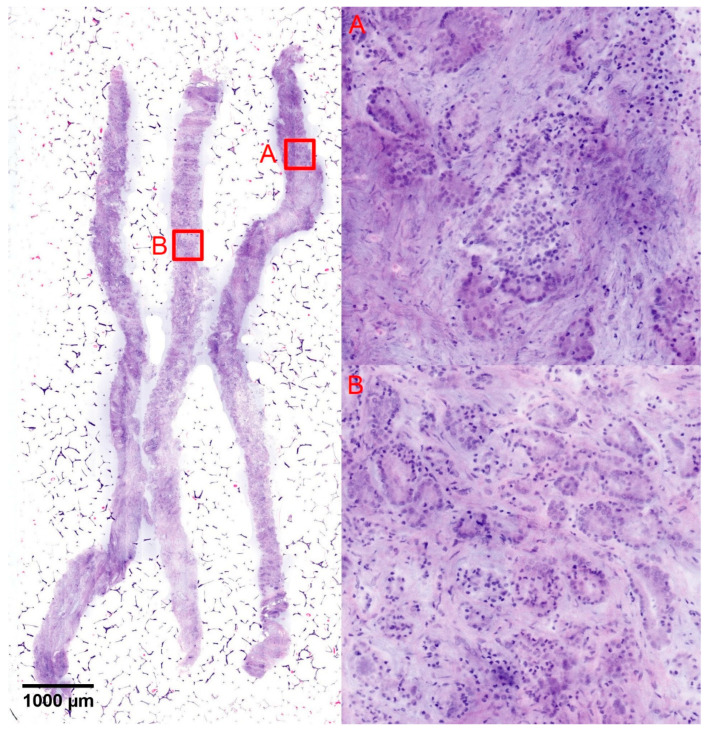
Diagnosis of GLEASON-4-Patterns in FCM-Scans. The figure shows the FCM scan of a biopsy with formations of a GS7a tumor. The GLEASON-4 patterns (**A**) may only be focal and difficult to detect within the surrounding acinar GLEASON-3 patterns (**B**). The correct classification depends on the experience of the examiner. Due to sampling errors, these discrete changes can be missed in FCM scans as well as in conventional histology.

**Figure 4 cancers-16-00873-f004:**
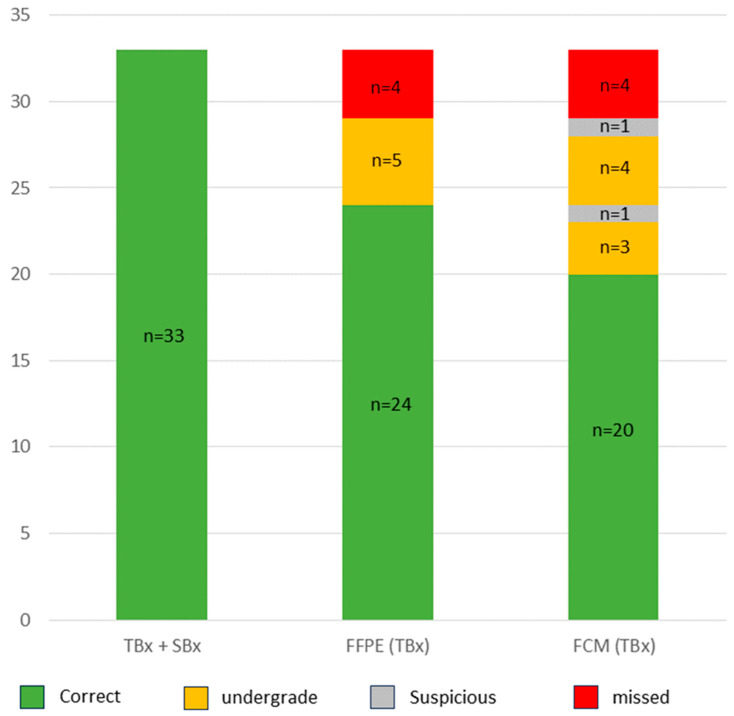
Detection rates of GLEASON-7a (GS3+4) Carcinoma in TBx und FCM. The cohort included 33 cases with GS7a carcinoma. Of these, 24 cases were recognizable by their high-grade component in the TBx. In 5 cases, only one GS6 component was detected in the TBx and 4 cases were not represented in the TBx, but only in the SBx. In the FCM, 20/24 GS7a carcinomas were detected, 3 cases were additionally under-graded and 1 case was only classified as tumor-suspect. All GS ≥ 7b tumors (n = 7) were recorded in the TBx and were reliably recorded in the FCM (not shown in the graph).

**Figure 5 cancers-16-00873-f005:**
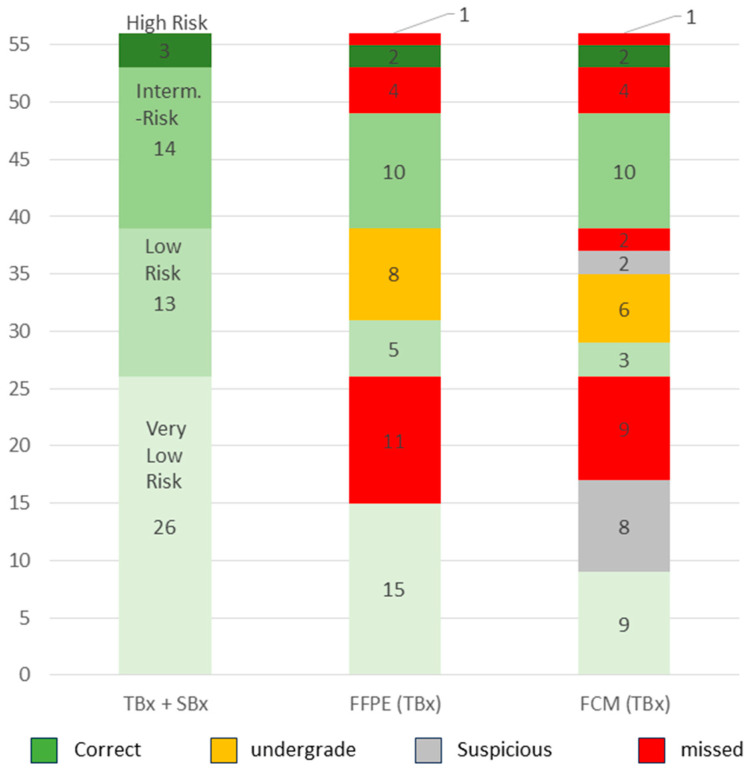
Detection rates and NCCN-Risk stratification of GS6-Carcinoma. The cohort included 56 GS6 carcinomas. The tumors in the very-low-risk group (n = 26) were not included in the TBx in almost half of the cases and were detected by FCM analysis in only one third of the cases. The majority of tumors in the low-risk group (n = 13) could only be correctly assessed in size together with the SBx and were therefore often under-graded as tumors in the very-low-risk group in the FCM analysis. The tumors of the intermediate-risk (n = 14) and high-risk (n = 3) group were classified consistently in FCM and histology.

**Table 1 cancers-16-00873-t001:** Comparison of the final histological results with the FCM diagnoses.

TBx + SBx		FCM_UT				FCM_BT			
Diagnosis	n	No Tumor	Suspect	GS ≤ 6	GS ≥ 7	No Tumor	Suspect	GS ≤ 6	GS ≥ 7
No Tumor	54	46	8	0	0	51	1	2	0
suspect	0	0	0	0	0	0	0	0	0
GS ≤ 6	56	16	10	30	0	23	4	29	0
GS ≥ 7	40	4	2	7	27	4	2	10	24
∑	150	66	20	37	27	78	7	41	24
%	100%	44%	13%	25%	18%	52%	5%	27%	16%
Sensitivity (GS ≥ 7)		68%				60%			
Specificity		100%				100%			
Positive p.v.		100%				100%			
Negative p.v.		89%				87%			
Kappa (GS ≥ 7)		0.75 (substantial)				0.69 (substantial)			
Kappa (overall)		0.55 (moderate)				0.54 (moderate)			
Diagnosis	n	No Tumor	suspect	VL Risk	L–H Risk	No Tumor	suspect	VL Risk	L–H Risk
No Tumor	54	46	8	0	0	51	1	1	1
suspect	0	0	0	0	0	0	0	0	0
VL Risk	26	9	8	9	0	16	0	7	3
L–H Risk	70	11	4	9	46	11	6	11	42
∑	150	66	20	18	46	78	7	19	46
%	100%	44%	13%	12%	31%	52%	5%	13%	31%
Sensitivity		66%				60%			
Specificity		100%				95%			
Positive p.v.		100%				91%			
Negative p.v.		77%				77%			
Kappa (L–H Risk)		0.67 (substantial)				0.56 (substantial)			
Kappa (overall)		0.52 (moderate)				0.49 (moderate)			

**Table 2 cancers-16-00873-t002:** Results of the intraday triage tests.

FCM of TBx		Histology (TBx + SBx)					
Diagnosis	n	No Tumor	Tumor	GS ≤ 6	GS ≥ 7	No Interv.	Therapy
No Tumor	66	46	20	16	4	62	4
Suspect	20	8	12	10	2	18	2
GS ≤ 6	37	0	37	30	7	30	7
GS ≥ 7	27	0	27	0	27	0	27
	150	54	96	56	40	110	40
FCM of TBx		NCCN-Risk stratification				Tumor Board	
Diagnosis	n	Very Low	Low	Intermediate	High	No interv.	Therapy
No Tumor	66	9	2	8	1	55	11
Suspect	20	8	2	2	0	16	4
VLR-Pca	18	9	6	3	0	9	9
L-HR Pca	46	0	3	31	12	0	46
	150	26	13	44	13	80	70

**Table 3 cancers-16-00873-t003:** Interobserver Agreement of the Results of the FCM-examinations.

		FCM_UT					
		No Tumor	Suspect	GS ≤ 6	GS ≥ 7		
FCM_BT	No Tumor	65	13	0	0	78	52%
	suspect	1	4	2	0	7	5%
	GS ≤ 6	0	3	35	3	41	27%
	GS ≥ 7	0	0	0	24	24	16%
		66	20	37	27	150	
		44%	13%	25%	18%		
Level of agreement (Κ)		0.78	substantial				

## Data Availability

The data are not publicly available due to privacy restrictions. The data presented in this study are available upon reasonable request from the corresponding author.

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
