# Peer review of "Ex Vivo Fluorescence Confocal Microscopy of MRI-Guided Targeted Prostate Biopsies for Rapid Detection of Clinically Significant Carcinomas—A Feasibility Study"

_cancers, 2024, doi:10.3390/cancers16050873_

Round 1

Reviewer 1 Report

Comments and Suggestions for Authors

This manuscript entitled “Ex Vivo Fluorescence Confocal Microscopy of MRI-guided targeted prostate biopsies for rapid detection of clinically significant carcinomas - a feasibility study” appears well-written. They reported an ex vivo fluorescence confocal microscopy examined specimen immediately from target biopsy. The result was good in sensitivity and excellent in specificity. Therefore, this method can be used for the assisting diagnosis of prostate cancer during MRI-fusion biopsy. It also can be used for helping focal therapy.

There is some spelling problem in the line 398, 399 and 400 such as fre-quently, al-lowed, etc.

Reviewer 2 Report

Comments and Suggestions for Authors

The author explored the utilization of microscopy technique which without the loss of patient biopsy material revealed pretty good diagnosis of prostate cancer that require treatment. Overall, it it’s a great study that will help in reducing diagnostic time, reduction in stress and will help in proper intervention by patients for their life style/psycho-therapy decisions. Some improvements are required in manuscript as below:

1.     Define PSA where it comes first in the manuscript.

2.     Define Prostate Imaging Reporting & Data System (PI-RADS) in introduction and its implications in terms of grades etc.

3.     Rename Figure4 as Figure1 everywhere.

4.     Section 2.5 on “Intraoperative FCM analysis of the target biopsies as a triage test” need to be expanded to include what are the features that are collected by FCM. What algorithm is used to make decision on the tumor state with what cut-offs/parameters?

5.     Line 272: : says “with 50%”: 50% of what? Please mention it.

6.     Discussion section has lots of words with “-“ in them. Please remove them. For example: al-lowed ; detec-tion ; fre-quently ; re-mains ;  intermedi-ate-risk ; addition-al ; partic-ularly ; char-acterize ; epithe-lia ; conven-tional ; depend-ing ; re-quiring ; cur-rently

Comments on the Quality of English Language

Lots of words with "-", they need to be removed.
